# Design to Achieve Accuracy in Ink-Jet Cylindrical Printing Machines

**Ivan Arango \*,† and Catherine Cifuentes †**

Department of Engineering, EAFIT University, Medellín 050022, Colombia; ccifuen1@eafit.edu.co
* Correspondence: iarango@eafit.edu.co; Tel.: +57-4-261-9500 (ext. 9084)
† These authors contributed equally to this work.

**Abstract:** Machines for direct digital inkjet printing on cylindrical containers are a new technology out on the market. The commercialization in the industrial sector has been affected by their high precision. This led to the use of mechanisms with narrow manufacturing tolerances and to the searches for topologies that have the least accumulated error without affecting quality. Machines with topologies that work on flat substrates have printing and productivity problems working on cylindrical substrates. This research paper presents the qualitative design of direct digital inkjet printers working over cylindrical substrates comparing five mechanical topologies; three topologies with radial distribution and two topologies with parallel distribution. The aim of these topologies is to find the precision, quality and efficiency of the printer taking into account the restrictions present in its construction. Each topology has separate constitutive mechanisms, the tolerance ranges between the movements of the print head and the substrate in order to determine precision are analyzed. Out of the five topologies described and analyzed in the phase diagram in section 3, three of them meet the requirements. One of the three topologies that meet the requirements is not being developed due to current technological limitations.

**Keywords:** inkjet; printer; topology; tolerance; machine

---

## 1. Introduction

Graphic arts printing is looking at penetrating a new market niche. For this reason, direct-to-object is an emerging technology which is achieving a performance that makes it a very competitive player in cost and versatility [1]. Research has focused on inkjet technology due to the connectivity with digital electronics which enables versatile result [2]. The advantages of this technology are supported in the non-use of ink transfer media and the versatility to change, but the quality and efficiency are still for small runs only [3]. The graphic information on the surface of the cylindrical containers is very important for the product image. Besides indicating the content, it is the publicity and the identity of the company that manufactures it. The food, chemical, pharmaceutical, beverage, cosmetics, and cleaning supplies industries use cylindrical containers for much of their production, and impact on the environment is key in this type of technology and would negate the real needs in product advertising [2,4]. Since its inception, inkjet printing has been developed with the premise that the surfaces on which it acts is flat, but recently, the direct impression is being developed on cylindrical containers and other solid ones with different forms and materials [5,6]. This is a task with a lot of challenges since inkjet printing for flat surfaces still does not match the speed and, in some cases, it does not match the quality of flexographic printing, either. For this reason, it is necessary to look for new topologies that assure the quality and productivity. In [5], Thorp makes a first approximation to the challenges that are in development for a prototype for printing on non-flat surfaces. It is argued that one of the greatest challenges in the development of the prototypes are the location

of the print heads against the solids, since this angle has a great influence on the alignment and, therefore, on the print quality [7]. The worldwide printer manufacturing industry consists of two segments of manufacturers: those that build all the constituent elements which are focused on large markets, and those that construct small batches or custom equipment using parts from specialized subsystem factories (Original Equipment Manufacturing OEM) [8–10]. The printing machines and their applications are developed according to the speed of the development of the technology that is given in the print heads [11,12]. The challenges of the innovation of these machines are being developed by small manufacturers. In [11], Cahill makes a detailed review of the main print heads available on the market for commercial printing machine manufacturers, which will be the starting point for mechanism constraints. When the print heads are not subject to the same structure or the substrate moves to each print head, it leads to an error position that is proportional to the print accuracy. The industrial printing machines accelerate and decelerate with high values but, at the same time, the mass in the slides are lights, then this combination gives a relatively low dynamic load. A low dynamic load makes the structure simpler and eliminates errors that arise in the different configurations of the printer. The nozzles in the inkjet technology are the heart of the quality, because the distance between the nozzles will be the distance of the drop on substrate of the container. If the distance changes (more or less), the image will be distorted, and the quality will be poor. That is why the mechanical precision is connected to this parameter. A print head within 96 and 384 nozzles per inch has a range in distance between nozzles of 38 and 250 microns. The distance between nozzles are the maximum error permitted in the mechanisms during any movement to avoid figure degradation and becomes the level of mechanical precision needed in the design [13]. During the machines' period of operation, the relative starting position of each container is called the index [14]. The indexing in machines where the topology requires the container to be released and then transported to the next print head is monitored so that the different colors Cyan, Magenta, Yellow and Black, with their english acronym (CMYK), are not superimposed. The fact that there is displacement of the container when a color is printed, is one weakness of the topologies compared to a planar substrate. The first machines to print on non-flat surfaces preserved the topology of flat machines. The result is effective if the travel distance of the drops are short, otherwise, problems may occur when wide 2D flat print heads on 3D surfaces present distortions due to the differential area [13,15]. In Figure 1, it can be seen how the impression of points spaced at equal distances, $\Delta X$ does not have an equivalence on the cylinder surface where this distance becomes a variable $\Delta S$, deforming the original. From this it is concluded that when the flat topology is used, the container must have the axis of rotation parallel to the longitudinal axis of the print head; otherwise, taking into account that the references of industrial print heads can reach up to 70 mm in length, the nozzles at the ends are very high above the container [5].

Cylindrical containers are made of different materials: polyethylene, polyester, polystyrene, PVC, acetate, glass (it is the most used for home and chemical industrial, pharmaceutical, veterinary, agro-industry) paper, paperboard, synthetic paper (it is the most used for offices and the recycling process). This diversification of the substrates to be printed requires that the factors involved in the process, such as surface tension, spreading and others, which affect print quality [16], be studied in the ink. In this article, the methodical design that analyzes, through a black box the flow of energy, matter and information, and all the subsystems needed to design a machine are detected by the method of the transparent box [17]. In Section 3, five possible topologies are presented, analyzing their configuration and their constitutive mechanisms. Section 4 compiles error in printing quality on cylindrical containers and the evaluation of the topologies. Section 5 presents the print time and the conclusions.

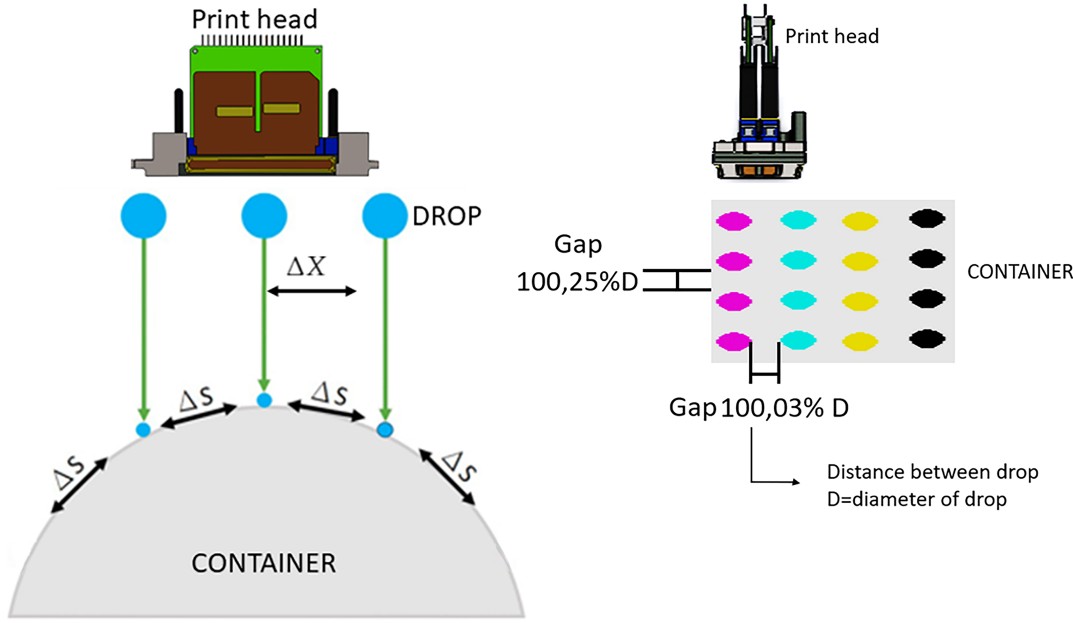

**Figure 1.** The 2D-3D Distortion and Drop Gap.

## 2. Methodology

As a first step in the designs, we analyzed if the existing machine can be adapted for the new products [18]. Designs of prototypes of printers for non-flat surfaces have been developed starting with Cartesian structures of two axes (sheet forward, transverse movement print head) for flat substrates [19]. In this topology, the paper feeder is replaced with the cylindrical container. Utilizing these same axis points for the paper feeder allows the rotation of the cylindrical container during the printing of non-flat surfaces. The printer approaches the desired result with the print head, but when using an industrial print head and a cylindrical container, the figure deforms due to the increase in the trajectory distance of the drop from the nozzle to the substrate (varies for all nozzles) then we start to construct Cartesian structures of four axes in which the print head is positioned in height and aligned with the axis of the container. The table is moved longitudinally and the fourth axis, which is mounted on the table, contributes to the rotation of the container.

The technique used in the investigation is the Methodical Designs [20] which is founded on the work of Pahl and Beitz [17]. The first step was a black box which described the input and output of energy flow, matter, and information of the whole system. In this case, as shown in Figure 2, print is the main function.

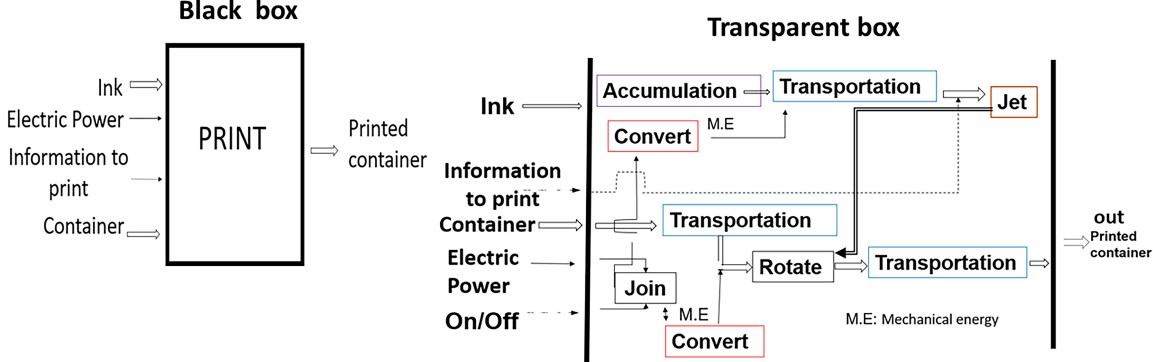

**Figure 2.** Black Box and Transparent box.

In the next step, the subsystems were identified. A block diagram consists of all the sub-functions separately identified by enclosing them in boxes and being linked together by their input and outputs

to satisfy the overall function of the printer machines. In other words, the original black box of the overall function is to redraw as a transparent box in which a sub-function is necessary and their links can be seen in Figure 2. An example of sub function is transportation.

Then, for each subsystem, several mechanisms, modules or devices that fulfill the respective function are searched. These are called the function or sub-system carriers by means of a functional structure.

Combining and permuting solutions of the main function are constructed using the sub-functions. Each solution is called a topology because it only includes functions that have a direct relationship with the location accuracy between the print heads and the container. Depending on the requirements, the commercial characteristics of the transparent box subsystems [17,21] are chosen. In this case, the transparent box consists of 7 sub-functions: feed–transfer–hold–rotate–eject–dry–eject which affect the container or print heads. The feeding and removal of containers from the printing area does not make any difference in the qualification, hence they are not included. For the analysis of tolerance of the parts composing the subsystems, it is assumed that low mechanical adjustment follows the norms set forth by ISO 286: 1988 or equivalent, ANSI B4.2-1978, EN 20286: 1993, JIS B 0401, DIN ISO 286, BS EN 20286, EN 20286 CSN.

According to tests by Mercier, et al., which use a print head with 760 nozzles, there is a pronounced decrease in print quality when the erroneous recording exceeds 50 microns [7,22]. The error in the register is given both in longitudinal and transverse displacement as well as in alignment or parallelism between the print head and the container [13]. To find the relative positioning error between each print head and the container (register), the cumulative error of the different mechanisms that compose each topology is evaluated. The tolerance field is used to evaluate the accumulated error of each topology. According to the International Committee for ISO Standardization [23], the tolerance required in precision machine parts is within the grades of IT6–IT7 for axes, and IT5–IT6 for holes in a range between 18 and 30 mm in diameter [23–25].

The following describes the types of errors that appear and influence the registry. Upon evaluating each topology, identifying the maximum error in the registry could affect the quality of the impression.

The first type of error is static and occurs when the mechanisms are left relatively misaligned or moved in the manufacturing or the set-up. To eliminate static errors in precision machines, the ones designed include static adjustment screws that allow the relative displacement of the mechanisms in a small length or angle. Static errors are not considered an accumulated error because they are eliminated as was previously mentioned.

The second type of error is a dynamic error. This happens when a mechanism moves, and by the effect of the tolerance in its construction, the drop is out of the printing area. The result is that between the print head and the container, there is longitudinal, transverse and misalignment displacement.

As the container moves through the printing process, the occurrence of an error due to the shift of the container or the print heads is called indexing. There are two ways this error can occur. Figure 3, illustrates how a rotation which is more or less than 360 degrees as the container passes from one print head to another may manifest itself. The error is usually attributed to the container shifting or skidding in the holding mechanism as it turns.

When the container has a length greater than the print length of the print head, the print head must print twice with displacement in order to cover the second pass that follows. Errors can occur in this displacement. In Figure 3, the displacement of the print head is observed when it surpasses the distance between the drops or it is superimposed in the impression.

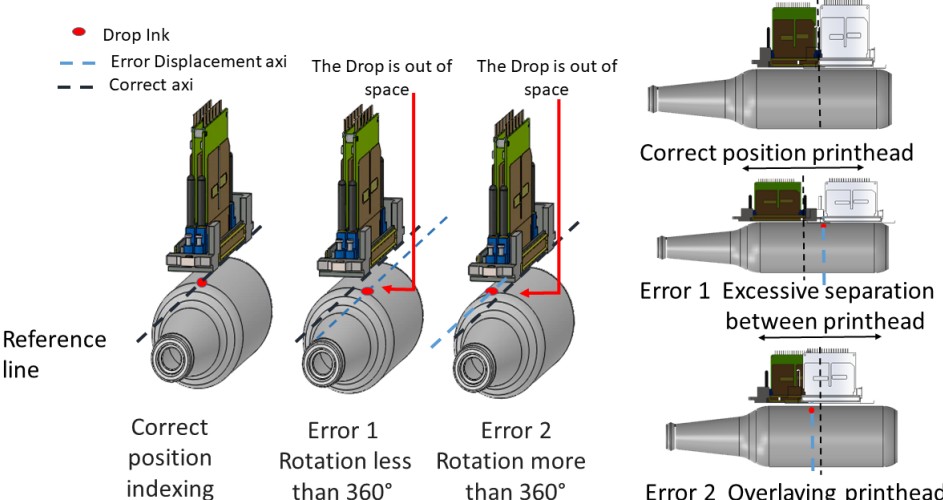

**Figure 3.** Print head cylinder rotation and translation error.

When a drop comes in contact with the surface, it tends to spread and penetrate the surface or spread. The spreading of the drop on the surface can (depending on parameters of ink and substrate) be irregular. [16,26]. Another phenomenon is called the drop break up, which occurs when the drop is broken during the trajectory from the head to the substrate [27]. The effect on the quality of the substrate is the mixture of colors. The main cause is the increase in the distance between the print head and the substrate. Other influencing factors are the drop velocity and the drop surface tension [28].

## 3. Topologies

The models can be classified into: concept machines, demonstration machines and productive equipment.

In concept machines for example (Profactor®, Heidelberg®) [29,30], presents a robot that works on sports shoes or on a ball printing machine. The time it takes is very high so it is only demonstrative. Another robot is presented by Xennia®, which takes pieces of form and places them under the heads in positions that are changing. The productivity is low.

In the demonstration machines, for example, the Advanced digital solution CP100® [31] presents a topology of advance in line and individual impression. Because the concept machine works only with one container and not with multiple containers, the performance is low.

For productive machines that are registered: the Tapematic® presents the CPrint mini machine which has an alternative conveyor belt that receives a container that is located under each print head and then ejects it and returns to take another container. The topology is similar to topology 1 with the difference that this topology works with multiple containers and the band run continuously. The print width is only one pass that corresponds to the print head width. The production is 800 containers per hour, but the container dimension is limited. Dobuit Machines® present the machine 9150 digital with topology and productive characteristics equal to the previous one, but doubles the size of the container in diameter from 50 to 100 mm.

### 3.1. Topology 1

In topology 1, Figure 4 all the print heads are mounted in a mono-structure (CMYK) or (CMYK + white background + transparent varnish) [32]. The mono-structure moves vertically to fit the diameter of the container and shifts depth to cover the entire length of the container. The separation distance between the print heads is greater than the maximum diameter of the containers. As seen in Figure 4, containers are printed simultaneously. The containers rotate in order to be printed and to advance longitudinally, which is to be located under each print head. In Figure 4 the phase diagram shows

each movement. The Translator is represented by the blue line, rotation is represented by the red line, the print head is represented by the green line and the movement of the print heads is represented by the purple line. The black arrows indicate when the container is held (up arrow) and released (down arrow). The *x*-axis is time and the *y*-axis is displacement.

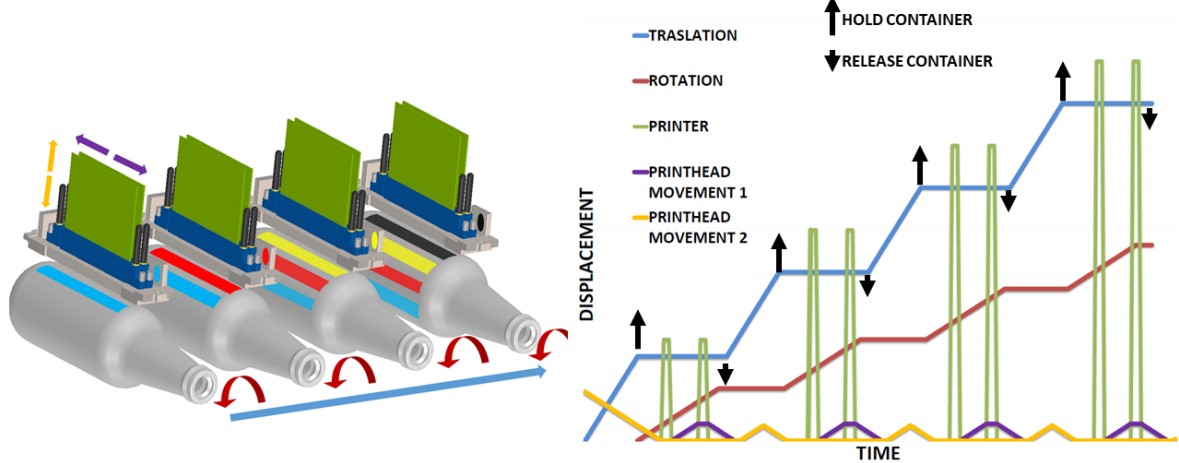

**Figure 4.** Topology phase diagram topology 1.

### 3.2. Topology 2

In topology 2, Figure 5 all the print heads are mounted in a linear monostructure (CMYK). The monostructure moves vertically to fit the diameter of the container. Four containers are printed simultaneously. Each spindle has the possibility of adjusting to a small distance (10 mm) in a longitudinal direction in order to synchronize with the starting index. In the case when the length of the container is longer than the print head, the containers rotate to be printed, and at the same time move longitudinally in the package as they are printed so that they can't stop. It obliges that software and the file management between print heads be different. This type of technology has not been developed yet. In Figure 5, it can be seen how the topology works.

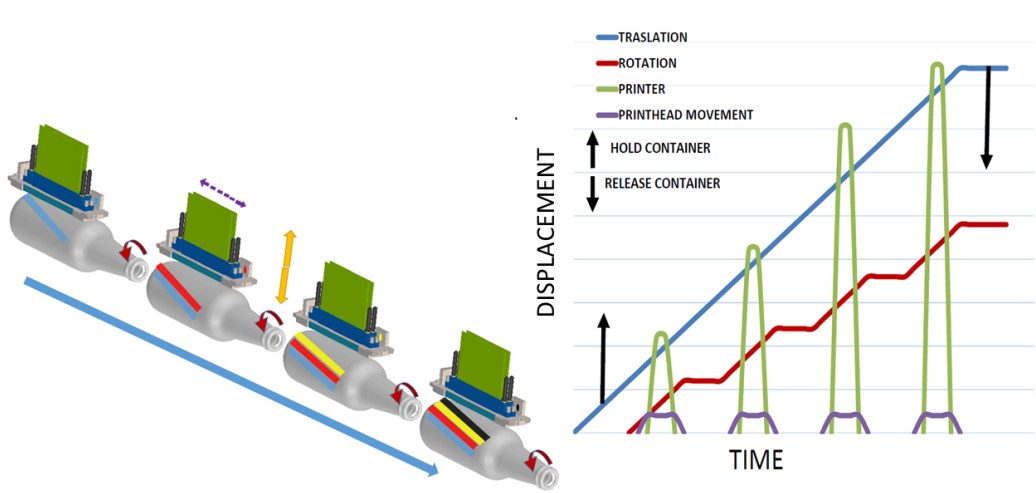

**Figure 5.** Topology phase diagram topology 2.

### 3.3. Topology 3

In topology 3, Figure 6, all the printheads are mounted radially in a mono-structure (CMYK). The height adjustment (diameter) of each print head is independent. The mono-structure is shifted in depth to cover the entire length of the container. This rotates in order to be printed. Although this topology initially presents advantages for the construction of small machines of prototyping, it is not enough, it has disadvantages because of its low productivity, it can print only one container at a time. Another disadvantage is the inclination of the print head. At this moment, there is not enough technology to guarantee the quality for printing. It is used only for the simplest printing like advertising prototypes of market evaluation, bar codes, and numbers, but the quantities and productivity are low.

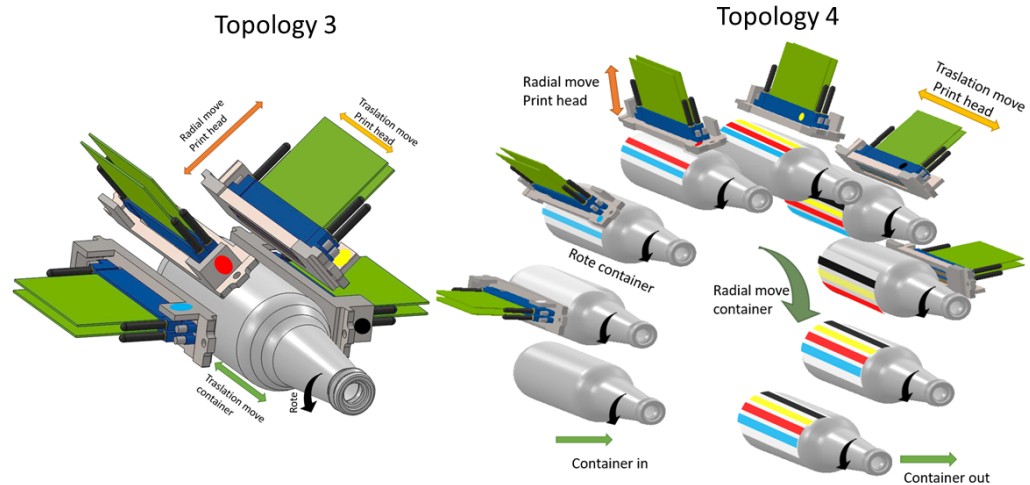

**Figure 6.** Topology 3 and 4.

### 3.4. Topology 4

In topology 4, Figure 6, as topology 3 the mono-structure (CMYK) and the height adjustment (diameter) are similar, however here four or more containers are printed simultaneously. The containers are mounted on a radial cylindrical turret. The container rotates on its own axis to be printed. Then the turret rotates and the container is then moved to place it below the next print head. The turret at the bottom has two stations that serve to enter and exit the containers. Although this topology initially presents advantages for the construction of machines of high productivity. The disadvantage is the inclination of the print head, in this moment does not have enough technology to guaranty the quality for printing. It is used only for simplest printing, like bar codes or numbers.

### 3.5. Topology 5

In topology 5, Figure 7 all the print heads are mounted radially in a mono-structure on the same plane (CMYK). The mono-structure moves vertically to fit the diameter of all containers. The print heads move individually in depth (radial) to cover the entire length of the container. Four or more containers are printed simultaneously. The container is mounted on a radial disc type turret. The container rotates on its own axis to be printed. Then the turret rotates and the containers are then moved in order to place them under the next print head. The turret has two stations that serve to enter and exit the containers. In Figure 7, it is evident how the topology works.

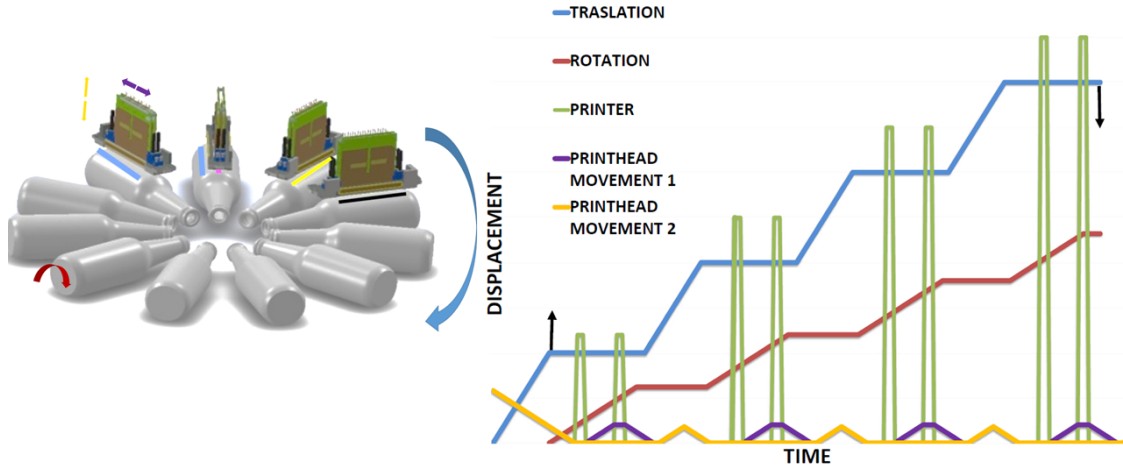

**Figure 7.** Topology 5.

## 4. Error in Printing Quality

### 4.1. Error in Printing Quality in Topology 1

In topology 1, a form to improve the print quality from to the print head is increasing the number of nozzles per unit length, but at the same time requires greater precision in the move's mechanisms. The precision in the manufacturing demands that specialized machines be used in the construction of the mechanism (grinding machines, machines of measurement by coordinates, lapping).

By selecting the topology that gets the best combination in precision, productivity and quality are analyzed. The magnitude of the error is found [25], which sums all the elements together.

$$\varepsilon_i = \sqrt[p]{\sum_{n=1}^{k_i} (|Error_n|)^p} \tag{1}$$

$\kappa_i$: Number of elements of the topology $i$
$\varepsilon_i$: accumulated error topology $i$
$Error_n$: element error type $n$
$p$: values near 1 for optimistic estimations and values near 2 for conservative estimations

This topology 1 (see Figure 8) is constituted by a functional structure, which begins with the feeding where the container is deposited in a support that is on a conveyor belt that moves it through the whole process. It then advances until it is under the first print head in the holding zone. Two conical surfaces are located in the extremes of the container. To get the container centered, the surfaces are moved against the container using a ball screw mechanism [6,24]. Once the container is held, the mechanism begins to rotate and the first print head begins to eject the ink. Once the first color is printed, the container is released and the conveyor belt is moved where the same printing process is repeated three times, but with a different print head color. Subsequently, the container is ejected from the printing line, ending the printed process Figure 8. The numbers in the figure are the different motors used to move the mechanism. Table 1 shows errors found in the analysis.

**Table 1.** Topologies Error [μ].

| | Motor 1 | Screw Ball Recirculating 1 | Motor 2 | Motor 3 | Band | Motor 4 | Screw Ball Recirculating 4 | Motor 5 | Screw Ball Recirculating 5 | Conical Surfaces | Total |
|---|---|---|---|---|---|---|---|---|---|---|---|
| **Topology 1** | | | | | | | | | | | |
| **Rotation container** | | | 5 | 5 | 100 | | | | | | 110 |
| **X container** | | | | 5 | 100 | | | | | | 105 |
| **Y container** | | | | | | 5 | 50 | | | | 55 |
| **Z container** | 5 | 50 | | | | | | 5 | 50 | 250 | 360 |
| **Topology 2** | | | | | | | | | | | |
| **Rotation container** | | | 5 | | | | | | | | 5 |
| **X container** | | | | 5 | 100 | | | | | | 105 |
| **Y container** | | | | | | 5 | 50 | | | | 55 |
| **Z container** | 5 | 50 | | | | | | | | | 55 |
| **Topology 5** | | | | | | | | | | | |

| | Motor 1 | Crown Gear 1 | Motor 2 | Motor 3 | Screw Ball Recirculating 3 | Motor 4 | Chuck Mechanism 4 | Crown Gear 2 | Cam Mechanism | —— | total |
|---|---|---|---|---|---|---|---|---|---|---|---|
| **Rotation container** | | | 5 | | | | | 100 | | | 105 |
| **angular traslation** | 5 | 100 | | | | | | | | | 105 |
| **Radial traslation** | | | | | | 5 | 50 | | 50 | | 105 |
| **Y container** | | | | 5 | 50 | | | | | | 55 |

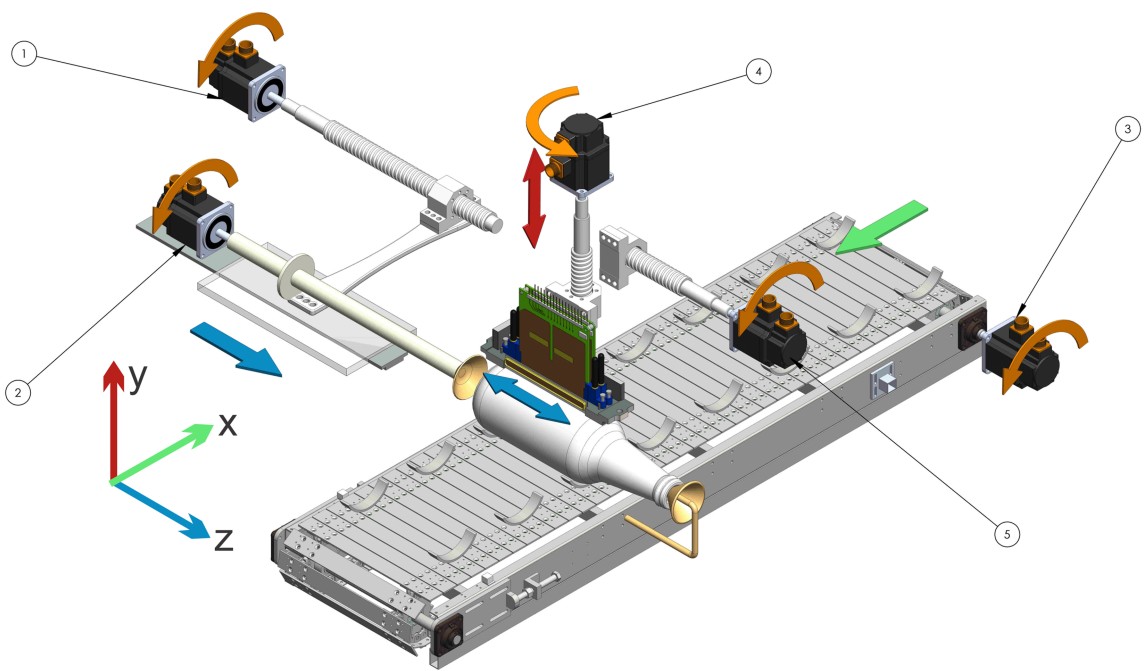

**Figure 8.** Scheme topology 1.

## 4.2. Error in Printing Quality in Topology 2

This topology 2 (see Figure 9) consists of a functional structure which consists of a support placed linearly one behind the other; when fed, a set of bars and wheels transmit the movement by turning the container continuously, while another transmission system performs the linear movement that would pass through all the print heads. When entering the container, this is held by means of supports and upper rollers which guarantee stability and concentricity throughout the system. The numbers in the figure are the different motors used to move the mechanism. Table 1, shows errors found in the analysis.

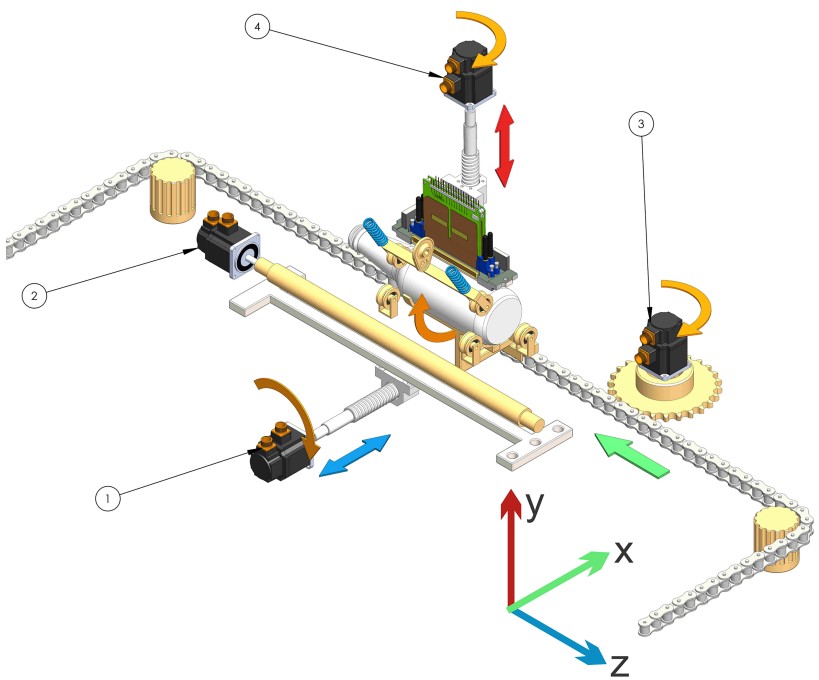

**Figure 9.** Scheme topology 2.

### 4.3. Error in Printing Quality in Topology 5

This topology 5 (see Figure 10) consists of a functional structure consisting of a shaft located vertically in the center of the main mechanism as a transmitter of the radial movement to the gears. In the lower part, there is a crown with internal bearings allowing the rotation of the mechanisms which are radially translating the container between the printheads. A gear facilitates the rotation of the container during a continuous printing process. During this process, the cam mechanism and spring hold and eject the container. The numbers in the figure are the different motors used to move the mechanism. Table 1, shows errors found in the analysis.

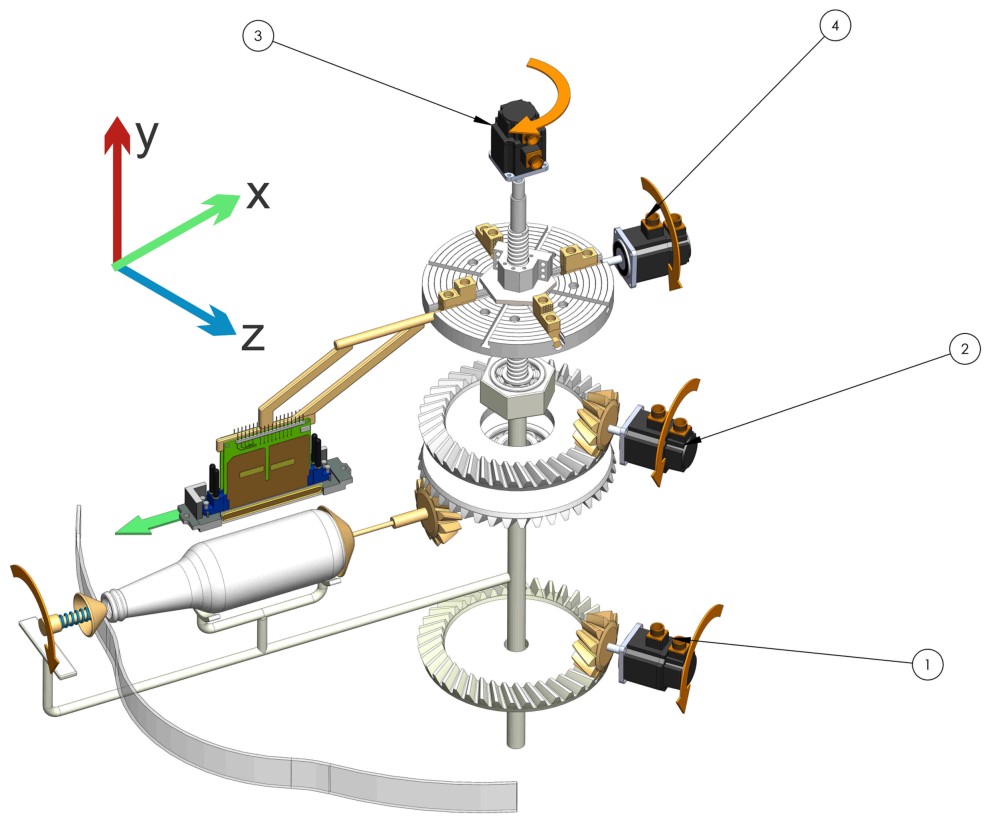

**Figure 10.** Scheme topology 5.

## 5. Print Time

The productivity of a printing machine is defined as the printed output per unit time. Inkjet technology in printing still does not compete in productivity with analog technologies (lithography, flexography, rotogravure, screen, dry offset), and from there comes the need to improve productivity [33] to perform comparable productivity with the analog technologies. The requirements needed to evaluate each stage of topology printing are relatively and parametrically indicated in Figure 11. The standard time is called the time that it takes to print a container. The times associated to other activities are assigned a number relative to the standard value as follows:

The overall process time is defined to the time that it takes for one container to go from one print head to another, thus, estimating the process time for one color:

process time for one color: "PT"

print head translation time: "phtt"

container traslation time: "ctt"

container rotation time: "crt" (note in this moment the print head is printing)

For topology 1 and 5

$$PT = ctt + crt + phtt + crt \qquad (2)$$

for topology 2

$$PT = crt + crt \qquad (3)$$

productivity = P Time unit = T

$$P = PT/T \qquad (4)$$

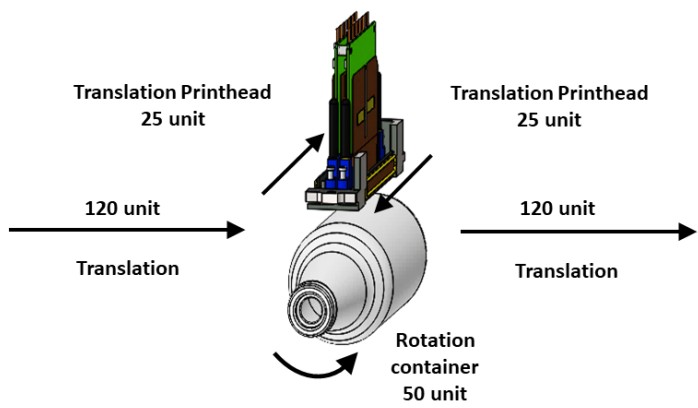

**Figure 11.** General print time process.

## 6. Conclusions

The qualitative design of a direct digital inkjet printer working over cylindrical substrates comparing five mechanical topologies was presented in this article; three topologies with radial distribution and two topologies with parallel distribution. The design criteria were precision as the principal criteria and productivity as the complementary criteria. The method used allows evaluable implements as options. The adaptation of a flat printer to cylinder printing does not give a positive result due to the perpendicularity between the axis of the container and the axis of the print head.

Cylindrical printing machines demand a high level of accuracy. That is why these are classified within a similar condition as light machine tools. For inkjet printing, the optimal topologies are oriented to machines with simultaneous printing because they can get closer to the high productivity of other printing technologies (for example flexographic). The main non-accuracy occurs when the container is released to be transported from one station to another, increasing the synchronization color error. One of the challenges in the design is to minimize the errors when the container is longer than the print head and one requires the movement of printing in the Z direction to be extended, including one more axis. Minimum errors were found in topology 2. Topology 5 was in the second place and topology 1 had the most amount of errors. The most productive machine, topology 2, has the lowest rating error during the displacement of the container between stations. Topology 2 qualifies as a high-efficiency machine but the software and file management between print heads need to be developed due to the figure in the file not being the same as the figure over the container. At the moment, the commercial offer of machines for inkjet printing is low and due to the new technology, the manufacturers' offer is low. As a final result, we considered focusing on building a detailed mathematical and 3D model to optimize the conceptual design as future work.

**Author Contributions:** All the authors contributed equal part to the work presented.

**Funding:** EAFIT University supported this research financially.

**Conflicts of Interest:** The authors declare no conflicts of interest.

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
