# Peer review of "Design to Achieve Accuracy in Ink-Jet Cylindrical Printing Machines"

_machines, doi:10.3390/machines7010006_

Reviewer 1 Report

You address an interesting topic. Nevertheless, the red line could be clearer and the findings should be discussed. The main parts are Table 1 and Chapter 3.9. I don't see any discussion of Table 1. It is still unclear to me. I'm not getting any information about the unit of the given numbers.

There is no discussing about print time in 3.9 - there are just giving some formulas. There is no value or indication given.

To give some specific comments:

Line 12-14: Noone knows about what topology 1..5 is like. Maybe a better description could be found.

Line 19: Missing dot and space at "... [1]The ..."

1. Introduction: You deal only with drop-on-demand inkjet technology. There is no indication. There is also no hint to CIJ (continous inkjet technology) given that could print also on larger distances.

Line 35: Missing space at "...[7].The ..."

Line 38: some strange thing at "[8], [9][10]."

Line 45: I cannot get the meaning

Line 46-49: message unclear and confuse

Line 52: Cyan

Line 58: "... [15] .In ..."

Figure 1: What does 100,25%D and 100,03% D mean? Explanation is missing.

Lines 65-67: Use () for additions in the enumeration

Line 69: What does "In this article, is developed, the ..." mean?

Line 79: "[18] .In ..."

Figure 2 right: There is "Transportation" several times given. Please double check if this scheme is logical.

Line 104: make any difference ...

Line 108: "which" wrong word; "with a 760 nozzles, ..."

Line 109: "... [22].The ..."

Line 125: "exakt position" > "defined position" would make more sense

Line 136-137: doublecheck sentence

3. Topologies: You mention profactor (reference is missing!) but do not mention heidelberg. This company has a commercial ball printing machine and also inkjet on robot

3. Topologies: You refer to topologies that are explained later. Try to resolve these references.

Line 176, 181: There must be a space betweenand

Lines 182-185: doublecheck sentence

Lines 192-194: on many robot arms inkjet heads are tilted    

Topology 4: There is no need to inclinate the printheads. They could all be directed vertically. Please take this fact into your considerations.

Line 215: Just the number of nozzles per unit length is not the ONLY quality parameter. A typical question is about ink-substrate interaction.

LIne 225: "p" shouldn't be bold

Line 227: double check sentence

Line 230: "Two ..."

Line 231: "[24] [6]. Once ..."

Line 238-241: double check sentence

Table 1: unclear. Any explanation is missing? Unit is missing?

Line 258: double check spelling

Line 258/259: double check sentence

Line 261: "T..."

Line 263: "P..."

Line 263: Printhead

Lines 263-264: Please order the content by somthing like ". , ; /"

Print time: is there any guess about time requirements?

Double check Lines 272, 279, 280

Line 285 ... for future ...

General:

For a submitted paper the spelling is too worse.

Please doubleckeck capital letters at descriptions of Figures

Is there any thought about how the ink is dried?

Author Response

Thanks for you review, it was very useful for us, in the .pdf file you can see the change and comments we did.

Reviewer 2 Report

The submitted work titled "Design To Achieve Accuracy in Ink-Jet Cylindrical Printing Machines" is novel and is an interesting work giving insights on the performance of the ink jet print output for a chosen topology (with either radial/parallel distribution).  The ink jet printing on a cylindrical platform o containers are new and yes, it is important to study the print output and quality for different design considerations that the paper has addressed.  The submitted work can be accepted in the present format after minor revision.  

1. Please check minor spelling errors and check the images and corresponding labels.

2. The author may need to rename the "discussion and conclusion" section to "conclusions".

Author Response

(The authors gave the same response as above.)

Reviewer 3 Report

The authors presented a very interesting study, machines for direct digital inkjet printing on cylindrical containers. The topic in the current study is very interesting because of the promising perspective to use inkjet printing not only for flat surfaces but also on non-flat/curved surfaces.

The following questions need to be address before publication:

Page 5, line 139, authors should expand & clarify that “another phenomenon is the drop break up” means.     

Page 6, there is a repetition of a paragraph: In topology 2, Figure 5 all the printheads are mounted in a linear monostructure (CMYK). The monostructure moves vertically to fit the diameter of the container. Four containers are printed simultaneously. Each spindle has the possibility of adjusting a small distance (10mm) in a longitudinal direction in order to synchronize with the starting index. Repetition of lines 174-177.

Page 7 repetition of lines 188-190 & 196-198. I suggest that the text repetition to be avoided. Alternative text can be used to describe the topology 4.  

Page 8 & figure 6 – this figure should be improved – very confusing/ difficult to follow & understand this image - topography 4. Also I suggest the following notation: Figure 6a for topography 3 and figure 6b for topography 4.

Figure 8 & 9 &10 should be improved – explain the meaning of component assigned with 1, 2..s.a. Overall, mismatch between figs 8-10, associated text and table 1.  

Some text corrections are necessary in special to avoid text repetition:

Page 8 line 216: precision

Page 9 line 220/221: which sums all elements together

Author Response

Thanks for you review, it was very useful for us, in the Word file you can see the change and comments we did.

Round  2

Reviewer 1 Report

I've worked directly in the PDF version of the document. Please find my comments in the attachement.

Author Response

Thanks for you  help. We finished to  revise the manuscript according to comments. we attach the pdf file with the corrections. the change were highlighted. 

kind regards
